# Clinical Impact of Prospective Whole Genome Sequencing in Sarcoma Patients

**DOI:** 10.3390/cancers14020436

**Published:** 2022-01-16

**Authors:** Luuk J. Schipper, Kim Monkhorst, Kris G. Samsom, Linda J.W. Bosch, Petur Snaebjornsson, Hester van Boven, Paul Roepman, Lizet E. van der Kolk, Winan J. van Houdt, Winette T.A. van der Graaf, Gerrit A. Meijer, Emile E. Voest

**Affiliations:** 1Department of Molecular Oncology, The Netherlands Cancer Institute, 1066 CX Amsterdam, The Netherlands; l.schipper@nki.nl; 2Oncode Institute, 3521 AL Utrecht, The Netherlands; 3Department of Pathology, The Netherlands Cancer Institute, 1066 CX Amsterdam, The Netherlands; k.monkhorst@nki.nl (K.M.); k.samsom@nki.nl (K.G.S.); l.bosch@nki.nl (L.J.W.B.); p.snaebjornsson@nki.nl (P.S.); h.v.boven@nki.nl (H.v.B.); g.meijer@nki.nl (G.A.M.); 4Hartwig Medical Foundation, 1098 XH Amsterdam, The Netherlands; p.roepman@hartwigmedicalfoundation.nl; 5Family Cancer Clinic, Netherlands Cancer Institute, 1066 CX Amsterdam, The Netherlands; l.vd.kolk@nki.nl; 6Department of Surgical Oncology, Netherlands Cancer Institute, 1066 CX Amsterdam, The Netherlands; w.v.houdt@nki.nl; 7Department of Medical Oncology, Netherlands Cancer Institute, 1066 CX Amsterdam, The Netherlands; w.vd.graaf@nki.nl

**Keywords:** whole genome sequencing, advanced sarcoma, diagnostic biomarkers, broad molecular profiling, precision oncology

## Abstract

**Simple Summary:**

Sarcomas are a heterogeneous group of diagnostically complex tumors with a poor prognosis and limited number of therapy options. Molecular profiling can aid pathological classification by detection of diagnostic biomarkers, and identify therapeutic opportunities for biomarker-based targeted treatment. Furthermore, pathogenic germline variants are present in ~10% of sarcoma patients, but remain often unrecognized. To explore the full spectrum of possible biomarkers in current molecular diagnostics, multiple and often iterative testing is required. In clinical practice, molecular profiling is selectively performed for specific patient groups with certain diagnoses already in mind. As a result, relevant diagnostic and/or actionable biomarkers are potentially overlooked. Whole genome sequencing (WGS) provides a complete, unbiased genomic characterization and detection of all possible genomic events within one diagnostic test. By applying prospective WGS in (suspected) advanced sarcoma patients in a tertiary sarcoma referral center, we uncover the missed potential of a targeted approach of molecular diagnostics.

**Abstract:**

With more than 70 different histological sarcoma subtypes, accurate classification can be challenging. Although characteristic genetic events can largely facilitate pathological assessment, large-scale molecular profiling generally is not part of regular diagnostic workflows for sarcoma patients. We hypothesized that whole genome sequencing (WGS) optimizes clinical care of sarcoma patients by detection of diagnostic and actionable genomic characteristics, and of underlying hereditary conditions. WGS of tumor and germline DNA was incorporated in the diagnostic work-up of 83 patients with a (presumed) sarcomas in a tertiary referral center. Clinical follow-up data were collected prospectively to assess impact of WGS on clinical decision making. In 12/83 patients (14%), the genomic profile led to revision of cancer diagnosis, with change of treatment plan in eight. All twelve patients had undergone multiple tissue retrieval procedures and immunohistopathological assessments by regional and expert pathologists prior to WGS analysis. Actionable biomarkers with therapeutic potential were identified for 30/83 patients. Pathogenic germline variants were present in seven patients. In conclusion, unbiased genomic characterization with WGS identifies genomic biomarkers with direct clinical implications for sarcoma patients. Given the diagnostic complexity and high unmet need for new treatment opportunities in sarcoma patients, WGS can be an important extension of the diagnostic arsenal of pathologists.

## 1. Introduction

Sarcomas account for ~1% of adult solid tumors. Although often grouped together based on their mesenchymal origin, sarcomas comprise a heterogeneous group of tumors, each with distinct pathological and clinical characteristics. With more than 70 histological subtypes, accurate pathological classification is challenging. In a series of 1463 second opinions of sarcoma patients, the pathological assessment of an expert pathologist did not match the initial diagnosis in almost half of the patients [1]. Evidently, accurate diagnosis is essential for clinical decision making, and misclassification may likely contribute significantly to suboptimal or even wrong treatment of sarcoma patients and bias in clinical studies.

Molecular profiling of sarcomas substantially aids pathological assessment, illustrated by several histological (sub)types that are characterized by diagnostic genomic events. For example, MDM2 amplification is present in most dedifferentiated liposarcomas and typically co-amplified with CDK4 and/or HMGA2, synovial sarcomas (SS) are often characterized by SS18–SSX fusions, and presence of MYOD1 mutations indicate spindle cell/sclerosing rhabdomyosarcoma [2,3]. While dedicated molecular assays are being used when specific diagnoses are being considered, extensive molecular profiling is not applied for all sarcoma patients. Immunohistological assessment alone results in misdiagnosis in 14% of sarcoma cases, also when the pathologist indicated to be confident on the initial diagnosis [4]. However, even with the opportunity to perform molecular profiling as part of the regular diagnostic workup, pathologists and oncologists are faced with the challenge to request the appropriate diagnostic test for the right patient. Consequently, diagnostic genomic events remain unnoticed, and unbiased whole-genome sequencing (WGS) led to diagnostic revisions in 3% of sarcoma cases that already received a full diagnostic workup [5].

In addition to supporting pathological classification, WGS can potentially further optimize clinical care of sarcoma patients, because germline DNA also is sequenced as part of the diagnostic workup. The use of paired germline and tumor DNA sequencing for somatic variant calling in WGS enables detection of previously unrecognized pathogenic germline variants. Germline variants in cancer associated genes are found in 10–15% of adult-onset sporadic sarcomas, but underlying hereditary conditions often remain unrecognized [6,7]. Furthermore, retrospective analyses on a large database of >3000 patients with metastatic cancer revealed that WGS can identify biomarkers with therapeutic implications for the majority of patients [8].

Hence, an unbiased complete genomic characterization holds the potential to prevent misdiagnosis, identify additional treatment options, and detect unrecognized hereditary conditions. Here, we describe the clinical utility of prospective WGS analysis in a series of 83 patients with advanced/metastasized sarcomas in a sarcoma expert center and demonstrate direct clinical implications for sarcoma patients.

## 2. Materials and Methods

### 2.1. Study Population

Patients were included as part of the WGS Implementation in standard Diagnostics for Each cancer patient (WIDE) study, in which 1200 patients with (suspicion of) metastatic cancer were included between April 2019 and January 2021. A detailed description of the study design and objectives can be found in the study protocol [9]. In short, WGS was performed in parallel with and independently of regular diagnostics for patients with advanced or (suspected) metastatic cancer that (a) underwent a tissue retrieval procedure, or (b) for whom a pathological assessment was requested on fresh-frozen archival tissue. The study was designed in concordance with the Declaration of Helsinki for medical research involving human subjects, Dutch law, and Good Clinical Practice and approved by the Medical Ethical Committee of the Netherlands Cancer Institute under registration number NL68609.031.18. Eighty-three patients with a (suspected) sarcoma at time of study inclusion were selected.

### 2.2. Sample Collection

Fresh tumor samples, either the primary tumor, a metastasis, or fluids containing tumor cells, were obtained during regular tissue retrieval procedures. After securing sufficient material for regular diagnostics, one sample was submitted for WGS analysis to the Hartwig Medical Foundation [9]. Additionally, a 10 mL blood sample was collected for sequencing of germline DNA to allow somatic variant calling compared to the patient’s own genetic background.

### 2.3. Regular Diagnostics

WGS analysis was performed independently of, and in parallel with, regular diagnostic analyses in the routine setting. Standardized, validated diagnostics tests were performed as requested by the treating physician or pathologist according to routine clinical care. The molecular diagnostic arsenal of the NKI department of pathology consists of targeted next generation sequencing (NGS) panel (Ampliseq, Cancer hotspot panel V2, Illumina Inc., San Diego, CA, USA), RNA-based NGS fusion analysis (Archer Fusionplex, Lung and Sarcoma panels, Archer DX Inc., Boulders, CO, USA), Sanger sequencing, reverse transcriptase polymerase chain reaction (RT-PCR), in situ hybridization (ISH) and immunohistochemistry (IHC). The pathology department operates under ISO15189 accreditation.

### 2.4. WGS Analysis and Reporting

We used standard procedures for WGS (Hartwig Medical Foundation, Amsterdam, The Netherlands), as previously described [10]. In brief, tumor DNA was isolated from fresh frozen tumor samples and sequenced at a depth of >90–100× coverage. Germline DNA from blood samples was sequenced at >30× coverage using the Illumina^®^ NovaSeqX platform. Comparison of germline and tumor DNA allowed for somatic variant calling, reporting all tumor-intrinsic genomic variants in the context of a patient’s own germline DNA. A detailed report with all genomic aberrations (variants, fusions, amplifications, structural variants, MSI, TMB and HRD) including their potential diagnostic and therapeutic relevance, was made available to the treating physician. Pathogenicity of variants was determined based on previously described oncogenic driver likelihood scores [8]. Where needed, pathogenicity of specific variants was further analyzed with additional diagnostic tests to determine gene and/or protein expression of potential pathogenic gene variants. For every variant, actionability was assessed in terms of regular therapy options (e.g., imatinib in KIT mutated GIST), or potential clinical trial allocation in ongoing trials within the Netherlands based on the detected biomarker. All WGS reports were discussed by a dedicated research board that consisted of a clinical molecular biologists, pathologists, clinical geneticists and medical oncologists before the results were disclosed to pathologists and treating physicians. Pathologists and treating physicians incorporated the genomic results within the clinical context. The bioinformatics for WGS analysis are publicly available [11]; HMF has ISO17025 accreditation.

### 2.5. Germline Findings

All patients were given an option for reporting of tumor associated germline variants at time of study entry (opt-in procedure). Germline findings were only reported when their presence had potential implications for tumor-directed therapy decisions and clinical follow-up of patients and/or their family members. In case of germline variant detection, patients were referred to the Department of Clinical Genetics for counseling. In addition, to assess the prevalence of pathogenic variants in our patient cohort, an anonymized, retrospective analysis of germline carrier status of pathogenic variants (class 4 or 5) in 49 cancer predisposition genes with diagnostic and/or therapeutic implications was performed (Appendix A) using the HMF pipeline [11].

### 2.6. Statistical Analysis

Descriptive statistics of patient and tumor characteristics and WGS results were generated. Numbers of tissue retrieval procedures and pathological assessment were compared between patients with and without diagnostic revisions using an independent sample *t*-test, using IBM SPSS v25 (SPSS, Chicago, IL, USA). Data on (duration of) response to any experimental treatment options identified by WGS analysis were not available.

## 3. Results

### 3.1. Characteristic Genomic Events

Eighty-three patients with a (suspected) sarcoma were included, consisting of 23 different histological tumor types (Table 1 and Figure 1). Thirty-two presumed sarcomas were characterized by a distinctive genomic event (Appendix A). Sixteen (rare) histological sarcoma subtypes were characterized by specific genomic events: a desmoplastic small-round-cell tumor harbored an EWSR1–WT1 fusion [12], an epithelioid hemangioendothelioma was characterized by a WWTR1–CAMTA1 fusion [13], an SS18–SSX1 fusion was detected in all three synovial sarcomas [14], a clear cell sarcoma had an EWSR1–ATF1 fusion [15], an NAB2–STAT6 fusion was present in two malignant solitary fibrous tumors [16], and an osseous spindle cell rhabdomyosarcoma was defined by an FUS–TFCP2 fusion [17]. Two other spindle cell/sclerosing rhabdomyosarcomas (SCSRMS) harbored MYOD1 p.Leu122Arg mutations [3]. All three myxoid/round-cell liposarcomas harbored a pathognomonic FUS–DDIT3 (previously known as FUS–CHOP) fusion [18]. Of the two endometrial stromal sarcomas (ESS), one was characterized by an MEAF6–PHF1 fusion [19]; no previously described fusion event was found in the other ESS. One of the two myoepithelial carcinomas of soft tissue showed an EWSR1–POU5F1 fusion [20]. In addition, well-differentiated and dedifferentiated liposarcomas (*n* = 10) were all characterized by MDM2/CDK4 co-amplification.

Four out of six gastrointestinal stromal tumors (GIST) had a KIT exon 11 deletion. Two patients, previously treated with imatinib, had secondary KIT mutations (p.Asn822Tyr and p.Val654Ala, respectively), which have been documented to confer imatinib resistance [21,22]. One patient was a previously unrecognized SDHA germline variant carrier and was diagnosed with a GIST. In the sixth GIST patient, WGS revealed an oncogenic NTRK driver mutation (p.Lys104del) that was confirmed with expression of the mutated allele in RNA-based analysis and positive NTRK immunohistochemistry. Extensive genetic testing in the routine (repetitive Sanger sequencing of KIT and BRAF, NGS Cancer Hotspot panel V2Plus and Archer FP Lung Target) had failed to detect a driver event.

Notably, almost all (presumed) leiomyosarcomas, i.e., 15 out of 17 in total, showed inactivation of both RB1 and TP53. Of the remaining two patients one had a RB1 wildtype and monoallelic TP53 mutated leiomyosarcoma. The other patient had a history of leiomyomatosis with progression to a low-grade intraabdominal leiomyosarcoma that had been previously resected [23]. Considering morphology and the RB1 and TP53 wildtype status, the intraabdominal biopsy was likely taken from a part of the leiomyomatosis.

In general, four genes (TP53, RB1, ATRX and CDKN2A) were altered in >10% of cases. In four sarcomas, no somatic driver events were detected. This included patients with leiomyomatosis, low-grade myofibroblastic sarcoma, osteosarcoma, and GIST. The patients with osteosarcoma and GIST had an underlying germline variant in TP53 and SDHA, respectively.

### 3.2. Diagnostic Revisions

By integrating complete genomic characterization into the diagnostic work-up, WGS led to a diagnostic revision in 14% of cases (12/83 patients, Table 2, Appendix A). One patient was referred to our hospital under the diagnosis of adenocarcinoma of unknown primary in the head and neck region with pulmonary metastases. After detection of an SS18–SSX1 fusion, it became apparent that the large epithelial component with HER2 overexpression of this tumor was misinterpreted as an adenocarcinoma, and the diagnosis was revised to a synovial sarcoma. For three patients, previously diagnosed with an alveolar rhabdomyosarcoma, embryonal rhabdomyosarcoma and undifferentiated pleiomorphic sarcoma (UPS) of the jaw, respectively, their diagnosis was revised to spindle cell/sclerosing rhabdomyosarcoma based on a MYOD1 p.Leu122Arg mutation in two patients, and an FUS–TFCP2 fusion in the presumed UPS patient. In another case, WGS revealed an EWSR1–POU5F1 fusion, changing the diagnosis from soft tissue Ewing sarcoma to soft tissue myoepithelial carcinoma. Finally, a thus far still unclassified tumor was recognized as a desmoplastic small-round-cell tumor after detection of an EWSR1–WT1 fusion.

Three suspected sarcomas turned out to be melanomas, a diagnostic pitfall that has long been recognized [24]. In all three cases, the tumor harbored genomic characteristics typical for melanoma: high mutational load compared to sarcoma samples, an ultraviolet-signature (COSMIC signature 7, C > T nucleotide changes), and typical driver events in the TERT promoter region (Figure 1) [25]. One of the patients also had an additional BRAF p.Val600Glu mutation. In more detail, one patient presented with an axillary lymph node metastasis of a presumed interdigitating dendritic cell sarcoma. A second patient was initially diagnosed with a schwannoma of the trigeminal nerve area that was resected. After rapid disease recurrence, the diagnosis had been revised to malignant peripheral nervous sheath tumor (MPNST) due to its aggressive clinical behavior, but the WGS analysis revealed a melanoma. Based on its morphology, deep dermal-subcutaneous localization, nerve infiltration, and lack of melanA/HMB45 staining, it was classified as a desmoplastic melanoma. A third patient presented with a presumed small-blue-round-cell tumor on his thigh (Appendix A), for which it was not possible to distinguish between melanoma and sarcoma due to the physical appearance and lack of distinctive immunohistochemical staining. It turned out to be a melanoma based on the genomic profile and presence of a BRAF p.Val600Glu mutation. Furthermore, a suspected locoregional recurrence of a dedifferentiated liposarcoma was reclassified as a secondary, radiotherapy-associated sarcoma based on the absence of MDM2/CDK4 co-amplification and widespread deletions with flanking microhomology, a signature that has been associated with ionizing radiation [26]. Finally, regular diagnostics could not resolve the differential diagnosis between pleural sarcomatoid mesothelioma and sarcoma. Absence of typical mesothelioma NF2 and BAP1 driver events argued against a mesothelioma [8], and subsequent negative H3K27me3 staining further strengthened the diagnosis of sarcoma.

Finally, one presumed wild type GIST turned out to be a KIT mutated GIST: a large KIT exon 11 deletion of 51 nucleotides was missed by NGS mutation analysis, but could be confirmed by Sanger Sequencing. Large deletions are challenging for regular NGS mutation analysis at the technical level (primer binding) and software recognition, which turned out to be missed with NGS analysis at two separate time points (Appendix A), and urged adaptations in the panel-based diagnostic workflow in case of a wild type GIST. To further explore the prevalence of large KIT alterations in GIST patients, we reviewed the WGS data of 71 GIST samples present in the HMF database. Large indels (≥47 nucleotides) were found in four samples (6%, Appendix A).

Further analysis of the patients’ medical history was performed to determine the number of tissue retrieval procedures and pathological assessments per patient (Table 1 and Appendix A). The diagnostic complexity of the revised cases was reflected by the high number of tissue retrieval procedures and pathological assessments of these cases prior to WGS analysis. All patients underwent multiple invasive procedures, biopsies and/or resections before the final diagnosis was established based on the genomic profile. In all cases, at least two different pathologists had assessed the case at separate time points in the diagnostic trajectory. With the exception of case 80, a presumed carcinoma of unknown primary, all cases were previously assessed by a pathologist in one of the Dutch sarcoma expert centers. For patients with a revised diagnosis after WGS, the number of prior tissue retrieval procedures and number of pathologists assessing the case to establish the final diagnosis was significantly higher than for patients whose pathological diagnosis did not change due to new findings with WGS (mean number of procedures: 4 vs. 2 (t (10) = 4.21, *p* < 0.05); mean number of pathological assessments: 6 vs. 2 (t (10) = 4.22, *p* < 0.05).

### 3.3. Germline Findings

In total, eight pathogenic germline variants were present in seven patients. Two germline variants, a BRCA1 p. Gln12* and a TP53 p. Arg196Ter mutation, were previously detected with regular germline diagnostics, and six pathogenic germline variants were not detected prior to WGS analysis. An SDHA p.Arg31* germline mutation was found in a young female patient with a wild type GIST, with no oncological family history, and consequently, no indication for regular germline diagnostics according to Dutch national guidelines. SDH immunohistochemistry was not yet performed at time of WGS analysis. SDHA has been considered as a GIST predisposition gene [27], and loss of succinate dehydrogenase was confirmed with immunohistochemistry. Secondly, a TP53 splice site mutation (c.782 + 1G > A) was identified in a male patient with a UPS and a history of Ewing sarcoma 26 years earlier. Finally, four germline variants were detected in CHEK2. These included an additional CHEK2 p.Glu64Lys variant in the BRCA1 mutated patient. CHEK2 was not included in the initial germline sequencing analyses, since at that time CHEK2 was not yet recognized as a (breast) cancer-associated gene. Furthermore, three CHEK2 c.1100delC variants, a known founder mutation in the Netherlands, were found. For one patient, there was a somatic loss of the second CHEK2 allele. The two other variants indicated CHEK2 carrier ship; there was no indication of biallelic CHEK2 loss in the tumor, and probably, these tumors originated independently of the CHEK2 germline variant. The clinical relevance of CHEK2 germline mutations in sarcoma patients is currently unknown [28].

### 3.4. WGS-Based Treatment Changes in Standard of Care and Additional Treatment Opportunities

Diagnostic revisions led to an adjusted regular treatment plan in seven patients, including a change of ongoing systemic treatment for two patients (Appendix A). Detection of a KIT mutation in a GIST led to an immediate change from sunitinib to imatinib, while retaining sunitinib as second-line therapy option. Next, a patient was receiving doxorubicin-ifosfamide chemotherapy based on a diagnosis of MPNST of the head and neck, but when the diagnosis was revised to desmoplastic melanoma based on WGS, the treatment was changed to immunotherapy according to melanoma guidelines. For a second melanoma patient treatment was adjusted by BRAF-directed (dabrafenib-trametinib) adjuvant treatment [29]. A young female patient was enrolled in a fertility preservation procedure to start subsequent intensive chemotherapy for a soft tissue Ewing sarcoma. However, after revision of diagnosis to a soft tissue myoepithelial carcinoma, the patient went directly for surgery without a need for (neo)-adjuvant chemotherapy. Another patient was scheduled for treatment with docetaxel-pertuzumab-trastuzumab for a presumed metastatic HER2-amplified adenocarcinoma in the head and neck region, after a potential HER2-overexpression on immunohistochemistry. After detecting an SS18–SSX1 fusion, the diagnosis was revised to synovial sarcoma and the therapy regimen was changed to doxorubicin-ifosfamide. A second patient with a presumed carcinoma of unknown primary was about to start with a CUP-directed chemotherapy regimen (gemcitabine-cisplatin) but received vincristine-ifosfamide-doxorubicin-etoposide instead after a diagnosis of desmoplastic small-round-cell tumor. Finally, doxorubicin-ifosfamide became a treatment option after diagnosis of sarcoma and exclusion of the diagnosis of sarcomatoid mesothelioma in a complex pleural tumor. In addition, one patient had been treated elsewhere in the past with methotrexate-doxorubicin-cisplatin for a then presumed osteosarcoma of the jaw that turned out to be spindle cell/sclerosing FUS-TFCP2 rhabdomyosarcoma of the bone, in hindsight a wrong treatment decision based on misclassification of the tumor.

As for experimental treatment opportunities for sarcoma patients (*n* = 80), WGS identified genomic biomarkers eligible for experimental targeted agents in 30 patients (Figure 2, 36%). In eight patients, more than one experimental therapeutic opportunity was identified. The majority of actionable events (26 biomarkers in 25 patients) were alterations in the RB pathway (CDKN2A loss or inactivating mutations, or CDK4 amplification), a potential target for CDK4/6 inhibitors. Other treatment options included PARP inhibitors for BRCA loss, CHEK2 loss, a CDK12 mutation, and an ATM mutation (*n* = 5), multikinase inhibitors for KDR-, PDGFRA-, and KIT amplifications (*n* = 4), an activating NTRK mutation (*n* = 1) and a NRG1 fusion (*n* = 1), checkpoint inhibitors for a tumor with a high mutational load, excluding the three melanomas, (*n* = 1), ALK inhibitors for an interstitial ALK deletion (*n* = 1), MEK inhibitor for a NRAS mutation (*n* = 1), and ERBB2/3 inhibitors for an ERBB3 amplification (*n* = 1). Eight patients started with a biomarker-based experimental treatment, with eleven more patients having an additional treatment option after progression on last line of regular treatment. As these patients were enrolled in ongoing clinical trials, information on (duration of) response to biomarker-driven experimental treatment is not available.

## 4. Discussion

Prospective use of WGS within the diagnostic workflow of sarcoma patients revealed genomic biomarkers with clinical implications in subsequent therapy decisions. This included 12 diagnostic revisions that led to a change of regular treatment plan in eight patients, 30 patients with actionable events, and six previously unrecognized germline carriers. These results are largely in line with the potential of broad molecular profiling in sarcoma patients that has been widely recognized previously. Actionable variants have been reported in 40–50% of sarcoma patients [30,31], germline variants are prevalent in 10–15% patients [7], and the utility of molecular profiling for accurate tumor classification has been established in previous clinical trials [4,5]. The extensive pathology testing to appropriately diagnose sarcoma patients is usually employed iteratively, while complete genomic characterization with WGS incorporates detection of all possible genomic events within one diagnostic test. Although most biomarkers that led to diagnostic revisions could have been detected with available standard of care molecular assays, the full spectrum of all possible biomarkers is only rarely explored in routine practice. All cases underwent multiple pathological assessments prior to the WGS analysis, during which the diagnostic biomarker remained undetected. Of note, this also included previously performed diagnostics in regional hospitals without specific sarcoma expertise prior to referral to the Netherlands Cancer Institute and evaluation by our institutional sarcoma expert pathologists. For only one case (patient 57), NGS-based fusion analysis was requested on the same sample as the WGS analysis, and the EWSR1–POU5F1 fusion and subsequent diagnostic revision were detected simultaneously with regular diagnostics and WGS analysis.

In addition to the recognition of all currently relevant genomic biomarkers, molecular profiling with WGS may improve our understanding and (future) treatment of sarcomas and may give a significant boost to future research. In this light, WGS holds several advantages compared to (large) panel sequencing. Firstly, newly discovered genomic biomarkers can be implemented directly within the WGS detection pipeline, without need for expansion of the diagnostic arsenal and indispensable validation of new biomarkers. The updated 2020 WHO classification on soft tissue and bone tumors reported a plethora of novel gene alterations for tumor classification, prognostication, and therapy decisions [32]. Within the rapidly expanding field of genome-driven sarcoma care, direct implementation of novel biomarkers is pivotal. Secondly, while four known sarcoma-associated pathogenic germline variants were detected, we also detected four CHEK2 germline variants with currently unknown relevance in sarcoma tumorigenesis [28]. However, detection of pathogenic germline variants in genes without conclusive sarcoma associations can have clinical implications for patients and their families and can further increase our understanding of cancer predisposition syndromes [33]. Thirdly, insights in tumor biology will undoubtedly improve tumor classification. For example, myxofibrosarcoma (MFS) has historically been regarded as a subset of UPS, until it was recognized that these tumors exhibit somewhat different clinicopathological features [34]. However, both tumor types are largely indistinguishable on a genomic level, and can still be regarded as a single disease entity on a phenotypic spectrum, suggesting that similar therapy approaches may be appropriate [2]. Hence, a better understanding of underlying tumor biology may support future interpretation of clinical trials. Fourthly, current cancer driver gene catalogues and large gene panel designs are primarily based on large-scale genomic cancer databases that are dominated by carcinomas [8,35,36]. By systematically collecting broad genomic information outside the currently recognized cancer-associated genes, we may identify novel sarcoma-specific driver genes in the future. Although outside the scope of this study, we confirmed the limitation of regular amplicon-based NGS analyses to detect large deletions in exon 11 of the KIT gene [37,38]. Given the frequency of large alterations in patients with metastatic GIST, broad genomic characterization including reassessment of KIT exon 11 could benefit GIST patients after a negative primary driver analysis.

We observed a higher proportion of diagnostic revisions compared to a previously reported revision rate of WGS in sarcoma patients (12 vs. 3%) [5]. This might be explained by an inadvertent patient selection due to the study design. All patients had advanced or metastatic disease, were adult patients, and were referred to the Netherlands Cancer Institute, a tertiary, sarcoma referral center. Additionally, our cohort consisted of several patients with well-established diagnostic pitfalls, including a synovial sarcoma with a large epithelial component that was misinterpreted as an adenocarcinoma [39], two melanomas with loss of immunohistochemical markers [24], a desmoplastic melanoma in the head and neck area that was classified as an atypical malignant peripheral nerve sheath tumor [40], and three patients with a revision in rhabdomyosarcoma subtype [41], of which one was originally diagnosed as pleomorphic sarcoma of bone in an expert sarcoma center. These cases might reflect a preselection in patients with above average complex tumors, since the treating surgical or medical oncologists most likely felt there was less need for WGS in more straightforward sarcoma cases. Of note, WGS is currently reimbursed for diagnostically complex tumors with uncertain diagnosis in the Netherlands [42]. Although these limitations hamper the generalizability of our results to the complete sarcoma population, these findings indicate that broad molecular profiling, in our case using WGS, is desirable for sarcoma patients with diagnostically complex tumors.

## 5. Conclusions

We demonstrated that WGS had direct clinical implications for 24% of sarcoma patients that were referred to a tertiary sarcoma referral center, by unraveling diagnostically complex sarcomas, identification of treatment options, and detection of unrecognized germline variants. We advocate systematic comprehensive genomic characterization of sarcoma patients in this setting to boost (future) research for these rare cancers.

## Figures and Tables

**Figure 1 cancers-14-00436-f001:**
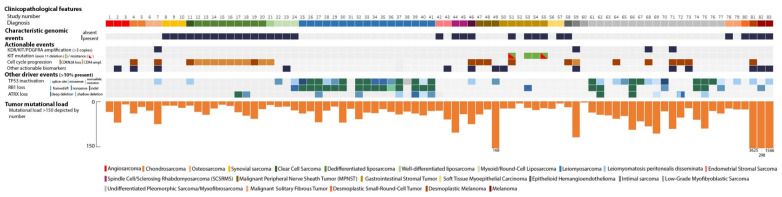
Oncoplot showing genomic information per sample.

**Figure 2 cancers-14-00436-f002:**
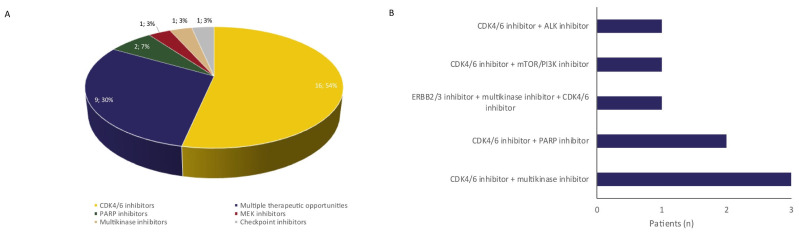
Experimental treatment opportunities identified with WGS. WGS identified experimental therapeutic opportunities for 30 patients (**A**), including eight patients with two experimental treatment opportunities (**B**).

**Table 1 cancers-14-00436-t001:** Patient characteristics.

Patient Characteristics	*n* = 83
Age at diagnosis, years	
Median	54
Range	22–83
Age at WGS analysis, years	
Median	58
Range	23–84
Gender, male:female	46:37
Previous lines systemic treatment, *n* (%)	
0	52 (63%)
1	18 (22%)
2	12 (14%)
3	1 (1%)
Primary tumor localization, *n* (%)	
Head/neck	8 (10%)
Intrathoracic/mediastinal	5 (6%)
Intraabdominal/retroperitoneal/pelvic	34 (41%)
Trunk	15 (18%)
Extremity	21 (25%)
Disease stage	
Metastatic	68 (82%)
Advanced	15 (18%)
Tissue retrieval procedures (*n*) *	
1	37
2	17
3	10
4	3
5	2
>5	2
Pathological assessments (*n*) *	
1	27
2	17
3	7
4	13
5	2
>5	5

* Number of tissue retrieval procedures and pathological assessments including revisions needed to reach final diagnosis. Data not available for 12 patients.

**Table 2 cancers-14-00436-t002:** Diagnostic revisions based on WGS analysis.

Study Nr	Suspected Diagnosis	Localization	Molecular Diagnostic Test Performed	Diagnostic Revision	Based on
8	Adenocarcinoma of unknown primary	Head/neck	IHC, SISH (HER2), NGS panel	Synovial sarcoma	SS18—SSX1 fusion
44	Alveolar rhabdomyosarcoma	Head/neck	IHC, RT-PCR, fusion analysis	Spindle-cell/sclerosing rhabdomyosarcoma	MYOD1 p.Leu122Arg
45	Embryonal rhabdomyosarcoma	Head/neck	IHC, FISH, methylation assay, fusion analysis	Spindle cell/sclerosing rhabdomyosarcoma	MYOD1 p.Leu122Arg
46	Osteosarcoma, undifferentiated pleiomorphic sarcoma of bone	Head/neck	IHC	Spindle cell/sclerosing rhabdomyosarcoma	FUS—TFCP2 fusion
47	Sarcomatoid mesothelioma vs. sarcoma	Intrathoracic/mediastinal	IHC, fusion analysis	Sarcoma NOS	Complete genomic profile, including lack of NF2 and BAP1 driver events
54	Wild-type GIST	Intra-abdominal	IHC, NGS panel (2×), fusion analysis	KIT mutated GIST	KIT exon 11 deletion (51 nucleotides)
57	Ewing sarcoma	Trunk	IHC, fusion analysis, RT-PCR (EWS1), FISH	Myoepithelial carcinoma	EWSR1—POU5F1 fusion
63	Dedifferentiated liposarcoma (recurrence)	Trunk	-	Radiotherapy-associated second primary	Lack of MDM2/CDK4 co-amplification
80	Carcinoma of unknown primary	Intra-abdominal	IHC	Desmoplastic Small-Round-Cell Tumor	EWSR1—WT1 fusion
81	Malignant peripheral nerve sheath tumor	Head/neck	IHC, fusion analysis	Melanoma	High ML/UV-signature, TERT promoter mutation
82	Melanoma vs. sarcoma	Extremity	IHC, fusion analysis	Melanoma	High ML/UV-signature, TERT promoter mutation
83	Interdigitating dendritic cell sarcoma	Extremity	IHC, NGS panel (2×), FISH	Melanoma	High ML/UV-signature, TERT promoter mutation

IHC = immunohistochemistry, SISH = silver in situ hybridization, FISH = fluorescence in situ hybridization, RT-PCR = reverse transcription polymerase chain reaction, NGS = next-generation sequencing. Morphology and information on IHC can be found in Appendix A. Detailed information on genes included in NGS panels and fusion analysis can be found in Appendix A.

## Data Availability

A detailed description of genomic alterations is provided as a Appendix A. Complete genomic data is available for researchers at Hartwig Medical Foundation in an access-controlled mechanism, meaning that scientific, legal, and ethical aspects of intended data usage and applications are assessed by an independent scientific and data access board. Researchers can apply for a data request at https://www.hartwigmedicalfoundation.nl/applying-for-data/ (accessed on 13 January 2022).

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
