# Peer review of "Clinical Impact of Prospective Whole Genome Sequencing in Sarcoma Patients"

_cancers, 2022, doi:10.3390/cancers14020436_

Round 1
Reviewer 1 Report
The authors should be complemented on this very well written paper on a relevant topic.
Data are presented on 83 patients consecutively referred to a tertiary referral center with (suspected) sarcoma in whom WGS was performed along side regular diagnostic workup. Data on diagnostic revisions based on WGS are presented, as well as germline findings and impact on treatment plans.
General concept comments
- As the authors state in the discussion section, there is inevitably a selection bias due to the study design. Some patients were probably referred from regional community hospitals to this tertiary sarcoma center, but some were probably referred from other tertiary sarcoma centers because of diagnostic difficulties (or hopes of a targeted treatment option). If available, referral data should be shown. (i.e. referral from another center of expertise or community hospital)
- In line with this, there is broad consensus on the need for pathological assessment by an expert pathologist. Therefore, it would be interesting to see rates of diagnostic revisions based on WGS as compared with the diagnoses made by the expert pathologists. The added diagnostic value of WGS should be assessed by a comparison to the diagnosis by the expert pathologists and not to the diagnosis by a non-expert pathologist in a community hospital. If available, please add these data (i.e. diagnosis in referring hospital (community vs other tertiary sarcoma center), diagnosis by expert pathologist study center, diagnosis based on WGS)
- Although the data are interesting and relevant, I feel that the conclusion is not supported by the data presented. Firstly, the % of patients in whom WGS has direct clinical implications is probably lower when an expert pathologist is consulted first. Secondly - with the exception of some - clinical relevance of the altered diagnosis or treatment plan remains largely unclear and targeted treatments are mostly available only in the setting of clinical trials. Therefore, the data do not support WGS in all patients referred to a tertiary sarcoma center but specific groups may be identified based on these data. (ie patients with diagnostically complex tumors, in the setting of clinical trials) and the conclusion should be adjusted.
Specific comments
- page 5/6 lines 202-206: it should be mentioned in the discussion that WGS is currently available in the Netherlands for most patients with unknown primary tumors.
- page 7 lines270-272: Was immunohistochemistry performed in this patient? Current guidelines do recommend IHC in WT GIST patients.
Reviewer 2 Report
In the manuscript entitled “Clinical impact of prospective whole genome sequencing in 2 sarcoma patients” the authors describe the use of whole genome sequencing for patients suspected of having metastatic sarcoma as a sub-group analysis of the larger WIDE study for patients with metastatic cancer. They demonstrate a surprisingly high likelihood of WGS identifying genetic changes with direct clinical relevance. Specifically, 14% of patients had findings that led to a revision in the cancer diagnosis and 7 (8%) whose treatment plans changed as a result of the WGS result. Germline mutations were also identified, although the clinical relevance of the WGS over other standard methods is less clear. Overall, this is a well written and well executed study that highlights an important problem in patients with rare cancers that are often difficult to diagnose.
Comments:
- This alluded to, but how would the authors suggest incorporating WGS into routine use, and for which patients would they recommend it? In the US, insurance companies do not reimburse for WGS and its widespread adoption is not feasible outside of a research setting.
- A comment on how WGS might add to whole exome sequencing or a large, targeted NGS panel, options which are less costly, would strengthen the manuscript
- The font on Figure 1 is impossible to read and should be enlarged.
- In supplemental table “Tissue retrieval procedures” the dates of individual should be removed as this is potentially identifiable and does not add to the analysis or interpretation of the results
Reviewer 3 Report
Schipper et al. present the benefits of WGS in sarcoma patients referred to a tertiary sarcoma center. The study is well designed and shows direct implications of molecular diagnostics for clinical decision-making.
I congratulate the authors to this interesting and thought-provoking study and support publication without further revision.
The study will especially attract intrest in the field of pathology and oncology.
